# Fabrication and Applications of Potentiometric Membrane Sensors Based on γ-Cyclodextrin and Calixarene as Ionophores for the Determination of a Histamine H1-Receptor Antagonist: Fexofenadine

**DOI:** 10.3390/polym15132808

**Published:** 2023-06-25

**Authors:** Haitham Alrabiah, Essam A. Ali, Rashad A. Alsalahi, Mohamed W. Attwa, Gamal A. E. Mostafa

**Affiliations:** Department of Pharmaceutical Chemistry, College of Pharmacy, King Saud University, P.O. Box 2457, Riyadh 11451, Saudi Arabia

**Keywords:** fexofenadine, supramolecular chemistry, potentiometry, sensors, drug formulation and quality control

## Abstract

Supramolecular fexofenadine sensors have been constructed. Although noncovalent intermolecular and intramolecular interactions, which are far weaker than covalent contacts, are the main focus of supramolecular chemistry, they can be used to create sensors with an exceptional affinity for a target analyte. The objective of the current research study is to adapt two PVC membrane sensors into an electrochemical approach for the dosage form determination of histamine H1-receptor antagonists: fexofenadine. The general performance characteristics of two new modified potentiometric membrane sensors responsive to fexofenadine hydrochloride were established. The technique was based on the employment of γ-cyclodextrin (CD) (sensor 1), 4-tert-butylcalix[8]arene (calixarene) (sensor 2) as an ionophore, potassium tetrakis (4-chlorophenyl) borate (KTpClPB) as an ion additive, and (*o*-NPOE) as a plasticizer for sensors 1 and 2. The sensors showed fast responses over a wide fexofenadine concentration range (1 × 10^−2^ to 4.5 (4.7) × 10^−6^ M), with detection limits of 1.3 × 10^−6^ M and 1.4 × 10^−6^ M for sensors 1 and 2, respectively, in the pH range of 2–8. The tested sensors exhibit the fexofenadine near-Nernstian cationic response at 56 and 58 mV/decade for sensors 1 and 2, respectively. The sensors exhibit good stability, fast response times, accuracy, precision, and longer life for fexofenadine. Throughout the day and between days, the sensors exhibit good recovery and low relative standard deviations. Fexofenadine in its pure, dose form has been identified with success using the modified sensors. The sensors were employed as end-point indications for the titration of fexofenadine with NaTPB.

## 1. Introduction

The wide variety of supramolecular ionophores and their use in chemical sensors have sparked interest in analytical chemistry in recent years [1,2]. The molecular recognition process for most macrocyclic hosts is based on the non-covalent trapping of analytes as guest molecules in the host cavity. Macrocycles are chemically stable, easy to functionalize, and can serve as receptors for a wide variety of analytes as guest molecules [3,4]. The cyclization of various motifs based on aryl groups coupled via short linkers can result in macrocycles with a hydrophobic inner portion and a hydrophilic outer part with a range of functions [5].

Cyclodextrin (CD), a cyclized glucose polysaccharide with α-1,4-linkage that can have variable cavity sizes, is one of the most often utilized macrocyclic hosts [5]. The inner portion of CDs is hydrophobic, whereas the outer portion contains hydroxyl moieties that aid in water solubility. While adamantane cyclodextrin is a typical detection of host–guest interactions, CDs can also connect to a diverse spectrum of nonpolar small molecule guests, with binding affinities ranging from 100 to 1000 M^−1^ [6].

Calix[n]arenes, another well-established family of macrocyclic supramolecular hosts, have been employed as a receptor for both small cations and anions [6]. Calixarenes can be synthesized by combining a p-substituted phenol, resorcinol, or pyrogallol with an aldehyde. Calixarenes conjugated with naphthylidine have been shown to detect amino acids including cysteine, histidine, aspartic acid, and glutamic acid [7]. Hamuro and colleagues demonstrated that calix[4]arenes can target a protein (cytochrome C) and block protein–protein interactions [8].

Supramolecular materials can be made using straightforward techniques, and they can be combined with other functional materials to create multi-component sensors. Currently, supramolecular components are capable of not only detecting target analytes based on known ligand affinity or unique host–guest interactions but may also be used for drug delivery [9,10], adsorbent materials [11], and sensor development [12,13,14]. This study discusses the electrochemical behavior of the host–guest complexes generated by the host, such as cyclodextrin and calixarene, as well as their applications employing electrochemical sensors (Figure 1).

Fexofenadine does not easily pass the blood–brain barrier compared to other second- and third-generation antihistamines, and as a result, it has a lower sedative effect than first-generation histamine receptor antagonists. It operates as an H1 receptor antagonist. Moreover, it has effects that block alpha1-adrenergic, anticholinergic, antidopaminergic, or beta-adrenergic receptors [15,16,17]. The chemical structure of fexofenadine HCl is 2-[4-[1-hydroxy-4-[4-[hydroxy(diphenyl)methyl]piperidin-1-yl]butyl]phenyl]-methylpropanoic acid hydrochloride (Figure 1).

Several analytical methods, such as spectrophotometry [18], spectrofluorometry [19], voltammetry [20], HPLC-UV [21], HPLC-fluorescence [22], and HPLC-MS [23], have been used to assess fexofenadine in the literature. Nevertheless, these methods involve more difficult handling, expensive chemicals, and complex equipment. There are numerous advantages to using PVC-membrane-based potentiometric sensors. Potentiometric methods based on the use of ion-selective electrodes emerged as an alternative due to intrinsic benefits over those methods, such as portability and operation, wide linear dynamic range, reasonably fast response, and rational selectivity [24,25].

The use of ion PVC sensors in which the plasticizer was sparingly doped with soluble fexofenadine salts such as fexofenadine-ammonium molybdate [26,27], fexofenadine-phosphomolybdate [28], fexofenadine-tetraphenylborate or tetraiodomurcurate [29], and fexofenadine-phosphotungstate, modified with Zn and Cu oxide nanoparticles [30], facilitated simpler potentiometric determination based on the exchange equilibrium with the sample solution, albeit at a poor selectivity trade-off [31]. As a result, the usage of ionophores such as γ-cyclodextrin and calixarene appears to have stronger sensor properties in terms of selectivity as well as a greater linear response.

Ionophores are complex agents with a lipophilic nature; the high lipophilicity ensures significant retention within the membrane, enhancing the stability of the response and their lifetime [32]. On the other hand, the ionophore’s chemical structure contains polar functional groups that are essential for recognizing analyte ions and enhancing selectivity [33]. As a result, using an ionophore increases its selectivity, stability, and durability.

Based on the previously reported aspects, this research focuses on the accurate determination of fexofenadine using potentiometric sensors based on γ-cyclodextrin and calixarene as the molecular recognition host. To the best of our knowledge, this is the first study that uses supramolecular chemistry as novel ionophore-modulated sensors for fexofenadine assays. The current study aims to use γ-cyclodextrin (sensor 1) and calixarene (sensor 2) as ionophores in the presence of potassium tetrakis (4-chlorophenyl) borate as an ion additive in the development of novel fexofenadine membrane sensors. The methods were then used to detect fexofenadine in bulk and dosage forms as quality control instruments. The examined sensors are unique in that they are simple, sensitive, selective, accurate, fast, precise, affordable, and have a broader dynamic range.

## 2. Experimental Section

### 2.1. Apparatus

A Thermo Fisher Scientific Orion pH/mV meter (model 330), Waltham, MA, USA, was used for all potentiometric measurements. It was equipped with fexofenadine sensors and an Orion reference electrode, Ag/AgCl (model 90-02), which contained 10% (*w*/*v*) potassium nitrate in the outer compartment. An Orion 81-02 glass pH electrode was used to monitor the pH.

### 2.2. Reagents and Materials

All chemicals used were of analytical grade. Throughout the entire experiment, double-distilled water was used. High-molecular-weight PVC powder, *o*-NPOE, dibutyl sebacate (DBS), dioctyl phthalate (DOP), and tetrahydrofuran (THF) with a purity of >99% were all made available by Aldrich Chemical Company (St. Louis, MO, USA). Fexofenadine was purchased by the Sigma Aldrich Corporation in (St. Louis, MO, USA). The following materials were purchased from BDH, compound Ltd.: γ-cyclodextrin, 4-tert-butylcalix[8]arene, and potassium tetrakis (4-chlorophenyl)borate. The 120 mg Telfast tablets were provided by the nearby drugstore. A suitable amount of fexofenadine was dissolved in water to make a 0.01 M stock solution. The fexofenadine working solution (1 × 10^−2^–1 × 10^−6^ M) was made by serially diluting the stock solution in double-distilled water. A pH 3.5 phthalate buffer solution was created using potassium dihydrogen phthalate (0.2 M) and hydrochloric acid (0.1 M).

### 2.3. Preparation of Fexofenadine Sensors

In a glass Petri dish, 190 mg of PVC powder, 3 mg of potassium tetrakis(4-chlorophenyl) borate, 10 mg of γ-cyclodextrin (sensor 1) or calixarene (sensor 2), and 0.350 mL of DBS, DOP, or NPOE were thoroughly mixed (5 cm in diameter). The component mixture was dissolved in about 4 mL of THF. The solvent was allowed to evaporate overnight while the sensing membranes were being created after the components had been thoroughly combined to construct sensors 1 and 2. A polyethylene tube (3 cm long, 8 mm id) was attached to PVC master membranes by sectioning them with a cork borer (10 mm in diameter) and using THF to glue them to it [34,35]. We used electrode bodies that we designed in our lab. These bodies were formed of a glass tube with a polyethylene tube attached to one end, and they were filled with an internal reference solution (equal volumes of a 1 × 10^−2^ M aqueous solution of fexofenadine and KCl). A reference electrode made of Ag/AgCl with an internal diameter of 1.0 mm was employed. The indicator electrode was preconditioned by soaking for two hours in a 1 × 10^−3^ M aqueous fexofenadine solution, and while not in use, it was kept in the same solution.

### 2.4. Procedure

The sensors were calibrated using the reference electrode and a fexofenadine PVC membrane sensor as the indication electrode by dipping them into a 100 mL measuring cell containing 9.0 mL of phthalate buffer (pH 3.5) (Figure 2). A 1.0 mL aliquot of fexofenadine solution was added while being continuously stirred, resulting in a final fexofenadine concentration range of 1 × 10^−2^ to 1 × 10^−6^ M. The potential was measured once it had reached a steady value (E, mV). The calibration curves were created by plotting the observed potentials against the log (fexofenadine) value. The pre-made plots were used to compute unknown fexofenadine concentrations.

### 2.5. Determination of Fexofenadine in Dosage Form

Ten Telfast 120 mg tablets were weighed, and then they were ground into a fine powder. A suitable amount of powder containing 120 mg of fexofenadine was added to a 100 mL beaker, along with water. The mixture was then sonicated for around 10 min, vortexed for five minutes, and filtered. The filtrate was then diluted to the desired volume with water. A 50 mL beaker was filled with a 10 mL aliquot of the aforementioned solution, and the pH was then brought down to 3.5 using a phthalate buffer. Fexofenadine sensors and an Orion Ag/AgCl double-junction reference electrode were used to detect the test solution’s potential, and the concentration was determined using a calibration graph made from standard fexofenadine solutions under identical experimental conditions.

### 2.6. Pharmacopeia Method

HPLC analysis was performed on a Waters HPLC (Milford, MA, USA) equipped with a 1500 series HPLC pump. The Empower Pro chromatography manager was used for data collection and analysis. Waters, USA, phenylsilyl silica gel (125 mm 4.6 mm internal diameter 5 m particle diameter) was used. The mobile phase was ultrasonically degassed after being filtered through a 0.45 µm Millipore system. A dual-wavelength UV detector (2489) and an autosampler (717 plus) were used.

The mobile phase was made up of 350 volumes of acetonitrile HPLC grade and 650 volumes of buffer solution, to which 3 volumes of triethylamine were added and mixed (350:650:3 *v*/*v*/*v*). The separation was carried out in the isocratic mode with a flow rate of 1.5 mL/min at room temperature. The detection wavelength was set to 220 nm, and the injection amount was set to 20 µL. The buffer solution was a mixture of 6.64 g of sodium dihydrogen phosphate monohydrate and 0.84 g of sodium perchlorate in HPLC-grade water, and the solution was adjusted to pH 2.0 with phosphoric acid and diluted to 1000 mL with deionized water.

### 2.7. Potentiometric Titration

The potentiometric titration was performed by titrating an unknown solution comprising 3 mL of fexofenadine with a standard solution of 0.001 M NaTPB as the titrant and the suggested sensors as indicator electrodes. Ruggedness: Throughout the day and between days, quality control samples were analyzed using two distinct operators and two different instruments.

## 3. Results and Discussion

Selectivity response works by creating an inclusion complex (host–guest interaction) between fexofenadine as guest, γ-cyclodextrin, and calixarene as the host. Via their lipophilic interior spaces and hydrophilic outside surfaces, fexofenadine molecules communicate with the host. These polar functional groups are essential for identifying analyte ions [32]. The development of the inclusion complex depends on noncovalent interactions such as hydrogen bonds, electrostatic interactions, van der Waals forces, dipole–dipole interactions, steric effects, and other dispersion forces [36,37,38]. The ability of the ionophore to remove ions selectively, which is thought to be more important than the binding strength of the formed complex, is one factor that influences the selectivity of ionophore-based membranes [33].

### 3.1. Effect of Additive

The addition of an ion-exchanger or lipophilic ion neutralized the charge of the complexes produced between the carrier and the analyte. As a result of this procedure, the produced membrane has improved analytical behaviors depending on whether the lipophilic ion is cationic or anionic depending on the type of analyte [39,40]. In this case study, adding KTpClPB (as a negative site) improved ion extraction and ensured the sensing membrane’s permanent selectivity, allowing the detection of cations (such as fexofenadine ions) by minimizing anionic interference and enhancing target analyte selectivity [34]. As a result, the addition improves the proposed PVC sensors’ sensitivity and selectivity toward the proposed analyte [39,40].

The ionophore-to-lipophilic ion(KTpCIPB) ratio was almost 1:1 molar ratio and various concentrations of lipophilic ion were investigated; the results are shown in Table 1. The addition of 6.05 µM of KTpCIPB was the optimized value that demonstrated the best performance of the suggested sensors. The response mechanism of the investigated sensors containing either γ-cyclodextrin or calixarene ionophore as sensing materials is based on carrier mechanisms or host–guest interaction, which shows strong affinity towards the fexofenadine with good selectivity and Nernstian response. A different amount of ion exchangers was tested on the effective response of the proposed sensors. Results are presented in Table 1. The best composition of the membrane was observed at 6.05 μM of the ion exchange and 7.7 µM of the ionophore. The investigated sensors show good response in comparison with the membrane containing only the ion exchanger (as indicated in Table 1).

### 3.2. Formation Constant

The formation constant was calculated using the following equation [41]:(1)$${\beta \mathrm{IL}}_{n}=(\frac{L_{T}-nR_{T}^{-}}{Z_{I}}{)}^{-n}exp(\frac{E_{MZ_{I}}F}{RT})$$
where *E*_M_ is calculated by subtracting the cell potential of an ionophore-free membrane from that of a sandwich membrane [42]; *L*_T_ is the total concentration of ionophore in the membrane section; *R*_T_ is the concentration of lipophilic ionic site additives; *n* is the ion-ionophore complex stoichiometry; and *R*, *T*, and *F* are the gas constant, absolute temperature, and Faraday constant, respectively. Ion I of fexofenadine has a charge of *Z*_I_. The formation constant was denoted by the *β*IL*_n_* or Log *β*IL*_n_* symbols. Sensors 1 (γ-CD) and 2 (calixarene)’s formation constants were calculated to be 23.43 and 53.46 or 1.36 and 1.73, represented as *β*IL*_n_* or Log *β*IL*_n_*, respectively. According to the findings, fexofenadine interacts with the carrier, as shown by the formation constant.

### 3.3. Effect of Plasticizers

To examine the performance of the plasticizer, fexofenadine sensors based on γ-CD or calixarene with various plasticizers, DOP, DBP, and NPOE, were evaluated. The plasticizer, a fluidizer that enables uniform dissolution and the diffusion mobility of the electroactive substance, is widely acknowledged as a vital component of the PVC membrane sensor and is substantially necessary for assisting ion-exchange through the PVC-based sensors. Table 2 provides a summary of the sensor’s behaviors in relation to various plasticizers. In response to various plasticizers, sensor 1 responded with values of 54 mV/decade (DOP), 52 mV/decade (DBP), and 56 mV/decade (*o*-NPOE), while sensor 2 responded with values of 56 mV/decade (DOP), 53 mV/decade (DBP), and 58 mV/decade (*o*-NPOE). Thus, *o*-NPOE was the best plasticizer for use in the construction of suggested sensors. As a result, *o*-NPOE was used in all potentiometric tests (Figure 2).

### 3.4. Interferences Studied

Investigations were carried out with respect to how inorganic ions affected the way fexofenadine sensors responded. In a phthalate buffer solution with a pH of 3.5, the separate solution method (SSM) [43] was used to evaluate selectivity coefficients in accordance with IUPAC guidelines. The following equation was used to determine the selectivity coefficient determined by the SSM (2):(2)$$\log K_{A,B}^{pot}=\frac{E_{B}-E_{A}}{S}+\left[1-\frac{Z_{A}}{Z_{B}}\right]\log a_{A}$$
where a_A_ is the activity of fexofenadine; Z_A_ and Z_B_ are the charges of fexofenadine and interfering species; S is the slope of the calibration graph (mV/concentration); E_A_ and E_B_ are the potential readings observed after 1 min of exposing the sensor to the same concentration of fexofenadine and interfering species (1 × 10^−3^ each) alternately. Selectivity was also investigated using a membrane containing only an ion exchanger, which revealed that in the case of a membrane containing only an ion exchanger, all examined ions interfere (Figure 3). The proposed membrane sensors, on the other hand, are devoid of interference (Table 3)

### 3.5. Effect of pH, Response Time, and Soaking Time

The prosed fexofenadine sensors were studied in fexofenadine solutions at different pH ranges. To ascertain the ideal pH level for the tested sensors, the membrane response at various fexofenadine concentrations was measured in a variety of pH ranges. A very diluted solution of HCl or NaOH was added to adjust the pH (0.1 mM). At various pH values, the potential was measured for two distinct concentrations of fexofenadine (1 × 10^−3^ and 1 × 10^−4^ M). Figure 4 shows that the electrode responses for sensors 1 and 2 were constant over the pH range of 2–4 [44], and the response was 56 ± 0.5 and 58 ± 0.5 mV/decade, respectively. It was found that the suitable working pH was 3.5. Therefore, the working pH was 3.5, which agreed with all reported potentiometric methods (all measurements were carried out within the pH range of 2–4) [26,27,28,29]. A decrease in the associated potential response resulted from the creation of an unprotonated species of fexofenadine at pH levels higher than 9 [44], which also caused the formation of a free fexofenadine base. The average response time [43] is the time it took from the addition of the sample to achieve a constant stable potential reading. Fexofenadine sensors had a response time of around 15 s over the concentration range of fexofenadine, after which the potential reading was constant. Figure 5 depicts the reaction timings of fexofenadine sensors. The sensor’s reproducibility throughout the day or from day to day was studied, and the potential reading was stable throughout the day or between days. The sensor had a lifetime of more than sixty days, the potential slope was generally stable (±1 mV/decade), and the electrodes produced very reliable data. For the suggested sensors, the influence of soaking times was investigated for 0.5, 1, 2, 3, 5, and 6 h, as well as overnight. It was revealed that 2 h is adequate time for the condition of the new sensor section. In the case of 2 h or more, the RSD% number was less than 3%.

### 3.6. Sensors Characteristics

According to the recommendations of the International Union of Pure and Applied Chemistry (IUPAC), the analytical characterization of the proposed sensors based on γ-CD and calixarene as an ionophore was carried out [43]. Table 4 provides information on the analytical features of the suggested sensors. According to Equation (3), the calibration curve displays the linearity of the constructed sensors, as indicated by a logarithmic relationship between voltage (mV) and concentration (M):(3)$$E\left(mV\right)=S\log\left[fexofenadine\right]+Intercept$$
where “*E* (*mV*)” stands for the electrode’s potential, “*S*” is the electrode’s slope (56 mV/decade and 58 mv/decade for sensors 1 and 2, respectively), and the intercept is 321 ± 0.5 mV and 279 ± 0.5 mV for sensors 1 and 2, respectively. The calibration graph’s linearity was maintained over concentrations of 1 × 10^−2^–4 × 10^−6^ or 1 × 10^−2^–4.7 × 10^−6^ M with respect to fexofenadine (Figure 6). Table 4 presents the characteristics of two modified sensors, γ-CD and calixarene, together with the effects they have on sensor performance, calibration range, and slope (mV/decade). The correlation coefficient (r^2^) is 0.998, with a response time of 15 s and an optimal pH range of 2–4. The LOD and LOQ were evaluated in accordance with IUPAC recommendations [43]. The LOQ was 3.3 of the LOD, which was 4.5 × 10^−6^ and 4.7 × 10^−6^ M for sensors 1 and 2, respectively. The LOD was estimated as the concentration of fexofenadine by extending two straight lines that were 1.3 × 10^−6^ and 1.4 × 10^−6^ M for sensors 1 and 2, respectively, in the linear section of the calibration graph. The reported technique has a wider calibration curve than the published method [22,27,28], lower detection than the published method [28], and exhibited near-Nernstian responses than non-Nernstian responses [28], and it is consistent with the reported results [26,27,28,29]. Results are presented in Table 5.

### 3.7. Accuracy and Precision

The inter-day (repeatability) examination of fexofenadine, five replicates at the LOQ range, was used to examine the precision and accuracy of the approach. RSD% and recovery% of the measured concentration, respectively, were used to express the precision and accuracy of the procedure. Moreover, intraday and daily repeatability were looked into. For sensors 1 and 2, the intra-day accuracy ranged from 97.5% to 99.5% and from 98.0% to 99.5%, respectively, while the inter-day accuracy ranged from 97% to 99.2% and from 97.5% to 99.5% for sensors 1 and 2. In contrast, with respect to sensors 1 and 2, the RSD% for the intra-day ranged from 2.1% to 2.8% and from 2.2% to 2.8%, respectively. Inert-day RSD% ranged from 2.4% to 3% for sensor 1 and from 2.3% to 3% for sensor 2. The results are demonstrated in Table 6. The results fall within the acceptable range of less than 3.0% (precision) and greater than 97%.

#### 3.7.1. Ruggedness

The ruggedness of the potentiometric method was evaluated by conducting the study with two separate analysts (operators) and utilizing several instruments on different days. RSD values of less than 3% were noted for repeated measurements conducted using two separate instruments and operators over the course of three different daytime times. The results demonstrated that the investigated method is capable of generating highly precise results (Table 5).

#### 3.7.2. Robustness

The adaptability of the experimental variables that influenced the potential response served to highlight the method’s robustness. The approach appears to be quite resilient based on the preliminary examination of the results under these different circumstances; however, the pH of measurements should be between 2 and 4.

### 3.8. Application

The ability of fexofenadine membrane sensors to detect fexofenadine in dosage forms was initially evaluated by examining the recovery of a precise quantity of pure fexofenadine in solutions. The developed membrane sensors (1 and 2) were used to analyze solutions containing 5–500 µg/mL of fexofenadine (in five replicates) for the direct determination of fexofenadine. The average recovery for employing sensors 1 and 2 was 98.41 ± 2.2 and 98.0 ± 2.15, respectively. In contrast, the RSD for sensors 1 and 2 was 1.8% to 2.8% and 1.8 to 2.7%, respectively. (Results are in Table 7).

The developed sensors were used as the last stage to evaluate fexofenadine in its dose form. In Table 8, the outcomes are displayed. The results of the study of fexofenadine in its dose form were compared to those obtained using the *British Pharmacopoeia* method [45] (Table 8). The results indicate that the sensors offer a high level of precision and accuracy, matching the *British Pharmacopoeia* method [45]. The pharmacopeia method and the suggested sensors’ accuracy were compared using |t|_test_ for P = 0.05 and n = 5, which produced |t|_test_ between 0.14 and 1.05. These results showed that the suggested sensors are accurate relative to the pharmacopeia approach (|t|_test_ = 2.13) because they were lower than the listed value [45]. Using a two-tailed F_test_, the accuracy of sensors and the pharmacopeia approach were compared. The range of F_test_ was 1.29 to 1.77, which is less than the tablets (F_test_ = 6.38), which contained a significant difference [46]. These results show that both approaches provide comparable accuracy. The assay of fexofenadine in its dosage form was carried out using the suggested sensors with good accuracy and precision. Table 6 presents the obtained results.

#### Application of Fexofenadine Sensors as the Indicator Electrode

The created electrodes have been tested as end-point indication electrodes for potentiometric drug titrations in conjunction with an Ag/AgCl reference electrode. According to the findings of the titration of fexofenadine with sodium tetraphenylborate (0.001 M, each) using sensors 1 and 2 (Figure 7), it is obvious that the drug reacts with Na-TPB in a molar ratio of 1:1. The symmetrical titration curves for sensors 1 and 2 both had a very distinct potential jump of roughly 250.0 mV, demonstrating the excellent sensitivity of membrane sensors.

## 4. Conclusions

The created fexofenadine-PVC membrane sensor disclosed in this paper provides an alternative to the more time-consuming, albeit generic, chromatographic method and other documented methods for determining fexofenadine in pharmaceutical formulations. A novel fexofenadine-selective electrode based on γ-CD or calixarene as an ionophore is proposed. The incorporation of γ-CD or calixarene (based on the host–guest identification approach) in membrane composition, together with a lipophilic anionic additive (KTpClPB), enables easy-to-build sensors with fast responses, good sensitivity down to the micromolar level, and long lifetime and high selectivity. In comparison to the *British Pharmacopoeia* approach, the fexofenadine selective membrane was successful in determining fexofenadine in its formulation. T_test_ and F_test_ confirmed that the results met the standards of the statistical analysis. The proposed sensor has been successfully used as a quality control tool to identify fexofenadine in bulk and formulation. As indication sensors, the sensors were utilized to potentiometrically titrate fexofenadine.

## Data Availability

All data in the manuscript are available from all authors.

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
