# Peer review of "Fabrication and Applications of Potentiometric Membrane Sensors Based on γ-Cyclodextrin and Calixarene as Ionophores for the Determination of a Histamine H1-Receptor Antagonist: Fexofenadine"

_polymers, 2023, doi:10.3390/polym15132808_

Round 1

Reviewer 1 Report

Authors have modified the manuscript, and definitely it was improved after the revision, but some minor issues are stil remaining. The tables 2-4 are not uniform, in table 2 and table 4 authors report data corresponding to sensor 1 and sensor 2 as columns, in table 2 (ex-Table 1 for which a suggestion  to replace columns with  lines was done), the data are remained in lines. This mixed representation os confusing. SSM is separate solution method, and not "separate  separation"! - see Table 3 (*-marked clarification below the table 3, it was already noted to authors but they did non change this typos). Moreover, authors must add the standard deviation value for all the slopes reported in tables 1,2,4, and indicate a number of repetitions . The standard deviation are provided for Intersept in Table 4, but not for slope! The caption f a new Figure 2 is not legible.  After this corrections paper can be published in Membranes.

Author Response

Reviewer 1

1.The tables 2-4 are not uniform

Answer

The tables 2-4 have been uniformed and the data are inserted in columns. 

  1. SSM is separate solution method, and not "separate separation"! - see Table 3 (*-marked clarification below the table 3, it was already noted to authors but they did nonchange this typos).

Answer

Marked clarification in Table 3 has been corrected (separate solution method).

3.Moreover, authors must add the standard deviation value for all the slopes reported in tables 1,2,4, and indicate a number of repetitions. The standard deviation is provided for Intercept in Table 4, but not for slope!

Answer

The standard deviation value for all the slopes reported in tables 1,2, and 4 has been inserted.

  1. The caption f a new Figure 2 is not legible.

Answer

The caption of Figure 2 has been modified to be legible.   

Reviewer 2 Report

Manuscript is improved and can be published in its present form.

Author Response

The response is OK 

Reviewer 3 Report

The manuscript reported the design of a potentiometric sensor for the selective detection of fexofenadine. The various parameters of pH, time response, reproducibility, and accuracy of sensors 1 and 2 were studied. The manuscript is well-written and organized. 

Please see the following comments: 

1- It is highly recommended to address the work novelty.

2- The figure 2 caption is overlapped on the Figure 1A labeling. Please check and revise.

3- The pH range is too wide from 2 to 8. It is highly recommended to specify the best detection pH for sensors 1 and 2.

4- The mode of action for the selective response of fexofenadine molecules is unclear. It is highly recommended to address this point. 

5- It is highly recommended to address sensors 1 and 2 differences and the mode of action.

The manuscript's English language is good

Author Response

Reviewer 3.

1- It is highly recommended to address the work novelty.

Answer

The novelty of the work has been addressed (P.2 lines 83-88).

 2- The figure 2 caption is overlapped on the Figure 1A labeling. Please check and revise.

Answer

Figure 2 caption has been revised.

3- The pH range is too wide from 2 to 8. It is highly recommended to specify the best detection pH for sensors 1 and 2.

Answer

Although fexofenadine has two ionization groups corresponding to the free carboxylic group on the side chain (pKa = 4.25) and the substituted ring nitrogen (pKa = 9.53), Therefore, the suitable working pH was a wide range, and the optimum pH for both sensors is 2-4, which agrees with the most published methods [26–29] and has been corrected in the manuscript. 

4- The mode of action for the selective response of fexofenadine molecules is unclear. It is highly recommended to address this point. 

Answer

Selectivity response of fexofenadine has been inserted in the manuscript (P. 7 Lines 188-197).  

5- It is highly recommended to address sensors 1 and 2 differences and the mode of action.

Answer

Both ionophores belong to the same class of cyclic compounds known as cavitands in host-guest chemistry (supramolecular chemistry). Therefore, both ionophores have the same mode of action, and the difference is related to different chemical structures (as stated in the introductions P1 and P2).

Round 2

Reviewer 3 Report

The authors have been raised most of the comments. So, I recommend accepting the manuscript in the current form

The English language is acceptable